# Learning Robust Medical Image Segmentation with Inductive Bias

**Shrajan Bhandary**[1]               SHRAJAN.BHANDARY@TUWIEN.AC.AT
**Dejan Kuhn**[2]             DEJAN.KOSTYSZYN@UNIKLINIK-FREIBURG.DE
**Zahra Babaiee**[1]                 ZAHRA.BABAIEE@TUWIEN.AC.AT
**Tobias Fechter**[2]            TOBIAS.FECHTER@UNIKLINIK-FREIBURG.DE
**Anca-Ligia Grosu**[2]            ANCA.GROSU@UNIKLINIK-FREIBURG.DE
**Radu Grosu**[1]                   RADU.GROSU@TUWIEN.AC.AT

[1] *Cyber-Physical Systems Research Unit, Technische Universität Wien, Austria*

[2] *Department of Radiation Oncology, University Medical Center Freiburg, Germany*

**Editors:** Accepted for publication at MIDL 2026

## Abstract

Despite the success of transformer-based and convolutional neural networks in 3D medical image segmentation, current architectures exhibit limited generalisation on small datasets and under distribution shifts, especially when high-quality examples are scarce for specific structures. We introduce IB-nnU-Nets, a family of U-Net variants augmented with inductively biased filters inspired by vertebrate visual processing. Starting from a 3D U-Net backbone, we insert two 3D residual components into the second encoder block that implement on- and off-centre-surround convolutions with fixed, pre-computed weights and act as complementary edge detectors. Across multiple organ and tumour segmentation tasks, we show that equipping state-of-the-art 3D U-Nets with an IB block improves accuracy and robustness, with the strongest gains in small-data and out-of-distribution settings. The framework and trained IB-nnU-Net models are publicly available.

**Keywords:** 3D segmentation, inductive bias, limited data, out-of-distribution robustness

## 1. Introduction

Manual segmentation of organs and tumours from medical images is essential for diagnosis, treatment planning, and disease monitoring (Goldenberg et al., 2019; Spohn et al., 2021; Singh et al., 2020; Antonelli et al., 2022; Isensee et al., 2021). However, manual delineation is time-consuming and subject to substantial interobserver variability (Rischke et al., 2013; Steenbergen et al., 2015). Deep learning offers powerful alternatives for automatic segmentation. Since AlexNet (Krizhevsky et al., 2012), numerous convolutional neural network (CNN) architectures have been explored (Singh et al., 2020), with U-Nets (Ronneberger et al., 2015; Çiçek et al., 2016) becoming the dominant choice across many benchmarks.

Despite this progress, U-Net variants remain sensitive to imaging heterogeneity, shape variability across patients, and low tissue contrast (Gillespie et al., 2020). Even state-of-the-art frameworks such as nnU-Net (Isensee et al., 2021) show limited generalisation when training data are scarce or when deployed on out-of-distribution acquisitions (Litjens et al., 2017; Isensee et al., 2021). Scaling up models or datasets (Liu et al., 2023; Ulrich et al., 2023) can help, but requires substantial computational resources and curated data that are often unavailable in clinical settings. Moreover, recent work has shown that architectural

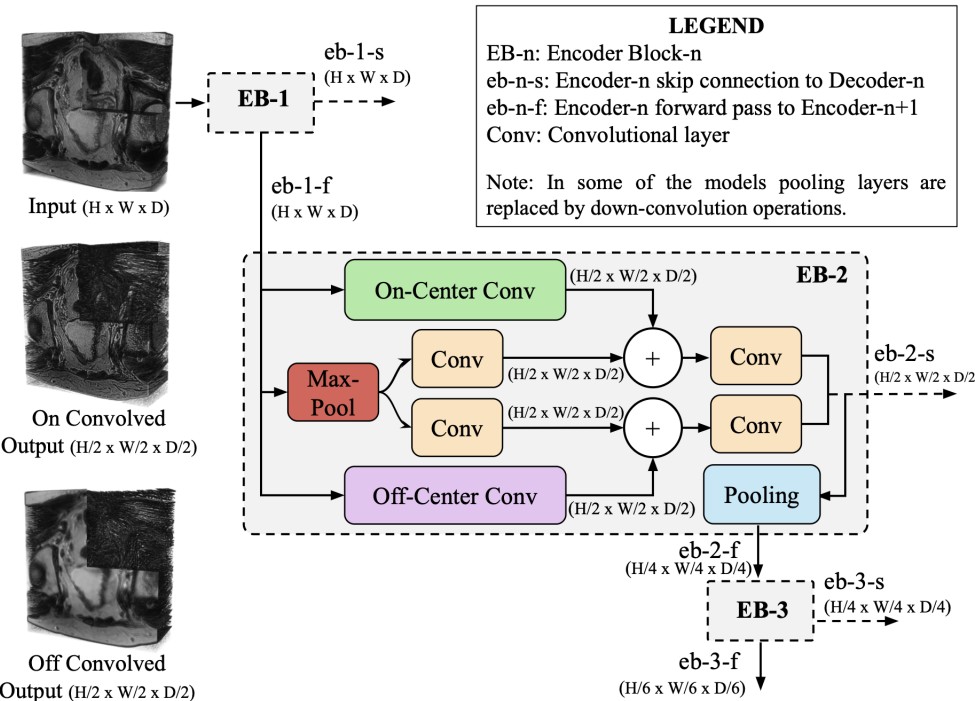

Figure 1: Extending U-Net variants with two 3D inductive-bias kernels (on- and off-centre-surround convolutions) in the second encoder block.

innovations yield diminishing returns on large datasets: properly configured U-Nets remain competitive with transformer-based alternatives under matched training budgets (Isensee et al., 2024). This motivates our focus on regimes where improvements are still achievable: small-data and out-of-distribution scenarios.

In this work, we introduce a biologically inspired inductive bias into U-Net architectures without increasing trainable parameters. Our approach draws on on- and off-centre-surround receptive fields in the vertebrate retina, modelled by difference-of-Gaussians (DoG) filters. We adapt previous 2D work on such inductive biases (Babaiee et al., 2021) to 3D, designing spherical kernels and integrating them into the encoder of 3D U-Nets. Concretely, we add two 3D residual components with fixed on- and off-centre-surround convolutions to the second encoder block (Figure 1), encouraging feature representations that emphasise edges and local contrast. We extend several architectures - Attention U-Net (Oktay et al., 2018), SegResNet (Myronenko, 2019), TransUNet (Chen et al., 2024), and nnU-Net (Isensee et al., 2021) - with these inductive-bias (IB) kernels and evaluate their performance on multiple organ and tumour segmentation tasks.

Our experiments show that IB-extended U-Nets are particularly beneficial in small-data and out-of-distribution scenarios, while maintaining strong performance on larger datasets. The 3D IB filters are modular, backbone-agnostic, and can be inserted into existing U-Net architectures without adding learnable parameters. In summary, our contributions are:

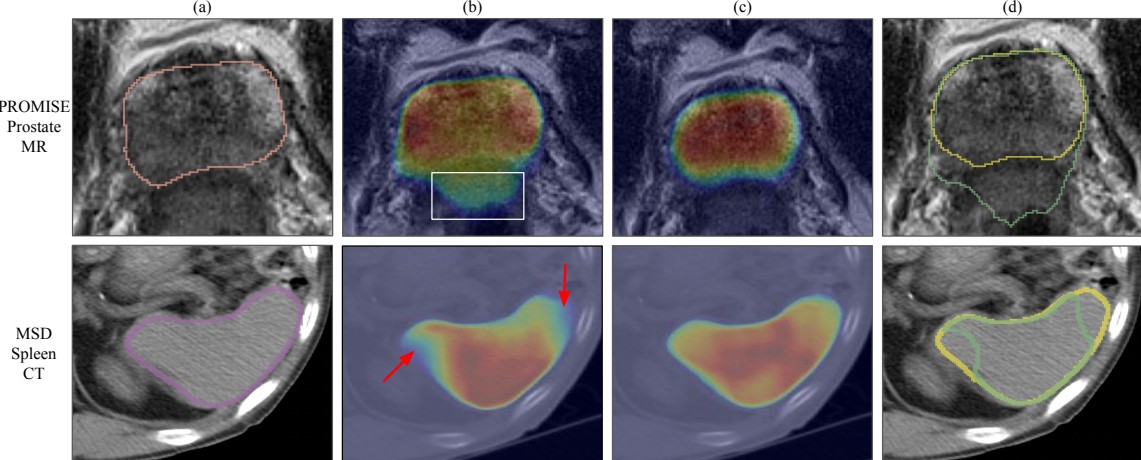

Figure 2: Qualitative comparison of attention maps for nnU-Net and IB-nnU-Net. (a) Raw MR and CT images with ground-truth annotations. (b) Attention maps of nnU-Net showing spurious and missed regions. (c) Attention maps of IB-nnU-Net showing robust segmentation performances. (d) Final predictions: green and yellow contours denote nnU-Net and IB-nnU-Net outputs, respectively.

- We introduce a 3D inductive bias based on spherical on/off centre-surround kernels and show how to integrate it into U-Net variants for 3D medical image segmentation, analysing kernel shape and encoder placement.

- We extend four architectures (nnU-Net, Attention U-Net, TransUNet, SegResNet) with this IB and demonstrate consistent robustness gains, with the strongest improvements in small-data and out-of-distribution settings.

## 2. Related Work

### 2.1. U-Net Variants

U-Nets (Ronneberger et al., 2015; Çiçek et al., 2016) and their derivatives form the backbone of most biomedical segmentation pipelines. They employ an encoder–decoder structure with skip connections that preserve spatial information and support high-resolution prediction. Attention U-Net (Oktay et al., 2018) introduces attention gates that emphasise relevant structures while suppressing irrelevant regions. SegResNet (Myronenko, 2019) replaces pooling with strided convolutions and adds residual connections. The nnU-Net framework (Isensee et al., 2021) is a self-configuring pipeline that adapts to each dataset and has achieved top performance in multiple segmentation challenges.

### 2.2. Medical Image Segmentation

Public datasets such as the Medical Segmentation Decathlon (MSD) (Simpson et al., 2019), Beyond the Cranial Vault (BTCV) (Gibson et al., 2018), and AMOS-2022 (Ji et al., 2022)

have accelerated progress in medical image analysis, with nnU-Net (Isensee et al., 2021) achieving top ranks on many benchmarks. Transformer-based models have recently gained traction: UNETR (Hatamizadeh et al., 2022b), Swin UNETR (Hatamizadeh et al., 2022a), and TransUNet (Chen et al., 2024) integrate transformer modules into the architectures.

However, Isensee et al. (2024) showed that many claims of transformer superiority over U-Nets do not hold under rigorous, controlled comparisons, with well-configured CNNs remaining state-of-the-art when training data and computational budgets are matched. Data efficiency and robustness therefore remain central challenges. For organ-specific segmentation, Bhandary et al. (2023) observed that robustness degrades substantially as dataset size decreases, underlining the need for methods that perform well with limited training data.

## 2.3. Biologically Inspired Architectures

Early vision models were heavily influenced by neuroscience and psychology, and biologically inspired ideas remain promising for artificial intelligence (Hassabis et al., 2017). Recent work revisits neuroscience-motivated modifications of CNNs to improve robustness and interpretability (Nayebi et al., 2018; Dapello et al., 2020; Babaiee et al., 2021). For example, Dapello et al. (2020) proposed a CNN architecture aligned with primate primary visual cortex that exhibits increased robustness to adversarial perturbations. Most relevant to our work, Babaiee et al. (2021) added 2D on/off centre-surround pathways, modelled by DoG filters, to CNNs and demonstrated improved robustness across several image classification benchmarks. We build on this foundation by extending the approach to 3D, designing spherical centre-surround kernels suited to volumetric medical images and adapting the integration strategy for U-Net-style segmentation architectures.

## 3. Materials and Methods

### 3.1. Datasets

We evaluate our approach on multiple public and private 3D datasets spanning different organs, modalities, and task difficulty levels. *AMOS-2022* is an abdominal CT dataset comprising 500 scans (300 training, 200 testing) with 15 organ annotations (Ji et al., 2022). We use the training split for 5-fold cross-validation and the spleen annotations from the held-out test set as an out-of-distribution target for models trained on MSD-spleen. *PROMISE-12* is a prostate MRI challenge dataset with 80 T2-weighted MR volumes (50 training, 30 testing) from multiple centres using heterogeneous acquisition protocols (Litjens et al., 2014b). The data exhibit substantial variability in voxel spacing, image quality, and prostate appearance. *MSD* refers to three organ segmentation tasks from the Medical Segmentation Decathlon (Simpson et al., 2019; Antonelli et al., 2021, 2022): MSD-hippocampus (394 volumes), MSD-prostate (48 multi-modal MRI scans), and MSD-spleen (61 CT scans). *PROSTATEx* comprises 204 prostate MRI studies acquired on two scanners (Litjens et al., 2014a). *Prostate158* is a multi-modal MRI dataset with 158 volumes (139 training, 19 testing) including prostate and tumour annotations (Bressem et al., 2022).

For tumour segmentation, along with the tumour subsets of the MSD dataset, we use two in-house PSMA-PET cohorts with different tracers: $^{68}$Ga-PSMA-11 (68 scans: 51 training,

17 test) and $^{18}$F-piflufolastat (65 scans: 45 training, 20 test). Annotations were obtained by expert consensus using fixed SUV windows tailored to each tracer ($^{68}$Ga: 0–5; $^{18}$F: 0–10).

### 3.2. U-Net Variants

We consider Attention U-Net (IB-Att-U-Net), SegResNet (IB-SegResNet), TransUNet (IB-TransUNet), and nnU-Net (IB-nnU-Net). These architectures represent diverse design choices - attention mechanisms, residual connections, CNN–transformer hybrids, and self-configuring pipelines - and are widely used in medical image segmentation. UNETR and Swin UNETR employ transformer encoders without convolutional backbones, making our fixed convolutional IB kernels difficult to integrate in a principled manner. We therefore focus on architectures with convolutional encoder blocks, where the IB can be inserted directly.

### 3.3. Inductive Biases in U-Nets

We combine biologically motivated filters (Babaiee et al., 2021) with U-Net variants to obtain more robust 3D architectures. This section describes the construction of our 3D IB kernels and how we integrate them into U-Nets.

#### 3.3.1. DESIGN OF THE 3D IB KERNELS

Retinal receptive fields in primates can be modelled by a difference of Gaussians (DoG) (Rodieck, 1965). For 2D kernels, a formulation by Kruizinga and Petkov (2000); Petkov and Visser (2005) defines the centre and surround weights as:

$$DoG_{\sigma,\rho}(x,y) = \frac{A_c}{\rho^2}e^{-\frac{x^2+y^2}{2\rho^2\sigma^2}} - A_s e^{-\frac{x^2+y^2}{2\sigma^2}} \tag{1}$$

where $\rho < 1$ is the ratio of the centre radius to the surround radius, $\sigma$ is the variance of the surround Gaussian, and $A_c$, $A_s$ are the centre and surround coefficients.

Naively extending this to 3D by stacking along the z-axis yields a cylindrical centre-surround structure, which is suboptimal for 3D medical images. Instead, we construct a spherical 3D centre-surround kernel:

$$DoG_{\sigma,\rho}(x,y,z) = \frac{A_c}{\rho^3}e^{-\frac{x^2+y^2+z^2}{2\rho^2\sigma^2}} - A_s e^{-\frac{x^2+y^2+z^2}{2\sigma^2}} \tag{2}$$

To balance excitation and inhibition while maintaining sufficiently large kernel weights, we enforce

$$\int [DoG_{\sigma,\rho}(x,y,z)]^+ dx\,dy\,dz = c, \quad \int [DoG_{\sigma,\rho}(x,y,z)]^- dx\,dy\,dz = -c \tag{3}$$

where $[w]^+ = \max(0,w)$ and $[w]^- = \min(0,w)$. In the continuous, infinite-support case, this implies $A_c = A_s$ (see appendix). For discrete kernels, we approximate the variance as:

$$\sigma \approx \frac{r}{\rho}\sqrt{\frac{1-\rho^2}{-6\ln\rho}} \tag{4}$$

| Comparison Type | Model Name/Setting | DSC ($\uparrow$) | HD-95 ($\downarrow$) | SDC ($\uparrow$) |
|---|---|---|---|---|
| Model variants | nnU-Net | 0.728 | 12.592 | 0.647 |
| | Cylindrical IB-nnU-Net (k=3) | 0.686 | 15.626 | 0.581 |
| | Cylindrical IB-nnU-Net (k=5) | 0.715 | 13.256 | 0.600 |
| | Spherical IB-nnU-Net (k=5) | **0.742** | **11.125** | **0.670** |
| IB filter placement location | Symmetric Encoder–Decoder | 0.706 | 13.830 | 0.616 |
| | IB filters in all encoders | 0.681 | 20.790 | 0.571 |
| | IB filters in only Encoder 2 | **0.742** | **11.125** | **0.670** |
| Different IB kernel parameters | $k=3$, $r=1$, $\rho=1/2$ | 0.711 | 15.949 | 0.638 |
| | $k=5$, $r=2$, $\rho=2/3$ | **0.742** | **11.125** | **0.670** |
| | $k=7$, $r=3$, $\rho=3/4$ | 0.738 | 11.581 | 0.659 |
| | $k=9$, $r=4$, $\rho=4/5$ | 0.740 | 11.362 | 0.660 |

Table 1: Ablation study on IB-nnU-Net design choices using a development subset of size 8 from PROMISE-12.

### 3.3.2. EXTENDING U-NET ARCHITECTURES WITH IB KERNELS

Using Equation (2), we compute fixed kernel weights for the On and Off pathways; the Off kernel is the sign-inverted version of the On kernel. For an input volume $\chi$, the on and off responses are obtained by convolving with the corresponding kernels:

$$\begin{aligned}
\chi_{\text{On}}[x,y,z] &= (\chi * DoG[r,\rho,c]^+)[x,y,z], \\
\chi_{\text{Off}}[x,y,z] &= (\chi * DoG[r,\rho,c]^-)[x,y,z].
\end{aligned} \tag{5}$$

To integrate these IB kernels into the second encoder block, we split its convolutional layers into two parallel pathways, each using half of the original filters, mirroring retinal on/off pathways. We add the 3D on and off responses to the activation maps of the first convolutional layer before max-pooling, using stride-2 IB convolutions to match downsampling; adding IBs after max-pooling yields inferior performance. We then concatenate the activation maps from the two pathways, producing an output with the same shape as the original block. Any 3D U-Net variant be extended to an IB-augmented version by replacing its second encoder block with an IB encoder block (Figure 1). This modification introduces no additional trainable parameters: IB-nnU-Net has identical parameter count to nnU-Net.

### 3.4. Implementation Details

All models were trained and evaluated using nnU-Net (Isensee et al., 2021). IB hyperparameters ($k = 5, r = 2, \rho = 2/3$) were chosen based on the properties of the spherical 3D kernels and validated on a small development subset ($n = 8$) from PROMISE-12, then fixed for all subsequent experiments. We trained all models from scratch under identical settings - same loss function, optimiser, learning rate schedule, augmentation, and hardware. We used combined cross-entropy and Dice loss, stochastic gradient descent with initial learning rate $10^{-2}$, a polynomial scheduler for 1000 epochs, and L2 regularisation ($10^{-5}$). Inference used sliding-window evaluation with 1/2 overlap. We report Dice similarity coefficient (DSC), surface Dice coefficient (SDC), and 95th percentile Hausdorff distance (HD-95). The IB kernels add no trainable parameters. VRAM usage increases by at most 1%, and training time by at most 2 seconds per epoch relative to the corresponding baseline U-Net. We use the Wilcoxon signed-rank test: IB-nnU-Net improvements with an asterisk (*) for $p \leq 0.05$.

| Size | Model | MSD-Hippocampus | | MSD-Prostate | | MSD-Spleen | |
|---|---|---|---|---|---|---|---|
| | | DSC(↑) | HD-95(↓) | DSC(↑) | HD-95(↓) | DSC(↑) | HD-95(↓) |
| 8 | SegResNet | 0.580 | 43.443 | 0.555 | 81.141 | 0.601 | 61.078 |
| | IB-SegResNet | 0.598 | 34.579 | 0.560 | 45.730 | 0.621 | 43.575 |
| | Attention U-Net | 0.644 | 21.012 | 0.456 | 94.860 | 0.615 | 51.639 |
| | IB-Att-U-Net | 0.652 | 20.283 | 0.657 | 15.365 | 0.630 | 33.087 |
| | TransUNet | 0.657 | 19.500 | 0.632 | 18.500 | 0.650 | 31.000 |
| | IB-TransUNet | 0.663 | 17.800 | 0.638 | 17.500 | 0.659 | 29.000 |
| | nnU-Net | 0.660 | 18.360 | 0.705 | 12.500 | 0.652 | 32.377 |
| | IB-nnU-Net | **0.670*** | **17.415*** | **0.720*** | **11.346*** | **0.665*** | **27.522*** |
| 16 | SegResNet | 0.694 | 14.744 | 0.705 | 27.255 | 0.704 | 16.513 |
| | IB-SegResNet | 0.709 | 13.574 | 0.754 | 13.992 | 0.715 | 14.331 |
| | Attention U-Net | 0.743 | 11.115 | 0.682 | 32.189 | 0.728 | 12.764 |
| | IB-Att-U-Net | 0.763 | 9.696 | 0.779 | 9.846 | 0.739 | 11.979 |
| | TransUNet | 0.750 | 9.730 | 0.782 | 9.900 | 0.745 | 10.400 |
| | IB-TransUNet | 0.810 | 9.250 | 0.793 | 9.800 | 0.747 | 10.300 |
| | nnU-Net | 0.762 | 9.763 | 0.804 | 9.714 | 0.752 | 10.274 |
| | IB-nnU-Net | **0.818*** | **9.177*** | **0.821*** | **9.672*** | **0.756*** | **10.181*** |
| 24 | SegResNet | 0.809 | 9.993 | 0.805 | 11.741 | 0.818 | 7.954 |
| | IB-SegResNet | 0.831 | 8.990 | 0.808 | 11.213 | 0.819 | 8.252 |
| | Attention U-Net | 0.851 | 8.660 | 0.816 | 10.972 | 0.841 | 9.725 |
| | IB-Att-U-Net | 0.862 | 8.604 | 0.819 | 10.106 | 0.843 | 8.920 |
| | TransUNet | 0.850 | 8.800 | 0.816 | 10.100 | 0.866 | 8.900 |
| | IB-TransUNet | 0.857 | 8.700 | 0.822 | 10.000 | 0.884 | 8.700 |
| | nnU-Net | 0.870 | 7.812 | 0.823 | 9.981 | 0.870 | 8.852 |
| | IB-nnU-Net | **0.879*** | **7.668*** | **0.831*** | **9.671*** | **0.888*** | **8.568*** |

Table 2: Accuracy and robustness of U-Net variants on MSD-hippocampus, MSD-prostate, and MSD-spleen for training subset sizes 8, 16, and 24.

## 4. Experiments and Results

We evaluate IB-extended U-Nets on challenging datasets, with emphasis on small training sets, noisy acquisitions, and out-of-distribution scenarios. First, we assess performance with limited training data by constructing subsets from MSD-hippocampus, MSD-prostate, MSD-spleen, and PROMISE-12. We randomly sample subsets of size 8, 16, and 24 from each training cohort and treat the remaining training images as test sets. Due to the small size of MSD-heart, we use subsets of 8 and 16. For PROMISE-12, we reserve a subset of size 8 as a development set for IB hyperparameter selection and ablations (Section 3.3).

Second, we evaluate performance using all available training volumes via 5-fold cross-validation on MSD-heart, MSD-hippocampus, MSD-prostate, MSD-spleen, PROMISE-12, AMOS-2022, and the two PSMA-PET datasets. In addition, we use AMOS-2022, PROSTA-TEx, and Prostate158 as out-of-distribution test sets. All experiments were conducted on an NVIDIA Titan RTX GPU (24 GB); adding IB kernels increased VRAM usage by less than 1% and training time by at most 2 seconds per epoch.

**Cylindrical versus Spherical IB Kernels:** Table 1 compares cylindrical and spherical IB kernels for IB-nnU-Net on a PROMISE-12 development subset. Spherical kernels substantially outperform cylindrical ones, whereas cylindrical IB variants can underperform the baseline nnU-Net, particularly when capturing 3D contours. Cylindrical kernels overemphasise planar structures and struggle with anisotropic voxel spacing and artefacts, supporting the necessity of the spherical 3D design.

| Size | Model | No noise | | Gaussian blur | | Random Gaussian noise | |
|---|---|---|---|---|---|---|---|
| | | DSC(↑) | HD-95(↓) | DSC(↑) | HD-95(↓) | DSC(↑) | HD-95(↓) |
| 8 | SegResNet | 0.559 | 55.413 | 0.392 | 127.450 | 0.510 | 90.016 |
| | IB-SegResNet | 0.624 | 29.484 | 0.495 | 66.931 | 0.639 | 22.567 |
| | Attention U-Net | 0.512 | 89.027 | 0.479 | 44.768 | 0.508 | 87.453 |
| | IB-Att-U-Net | 0.674 | 14.007 | 0.623 | 26.420 | 0.662 | 15.149 |
| | TransUNet | 0.722 | 13.200 | 0.701 | 15.462 | 0.716 | 13.560 |
| | IB-TransUNet | 0.730 | 12.400 | 0.713 | 14.167 | 0.718 | 13.464 |
| | nnU-Net | 0.728 | 12.592 | 0.707 | 14.750 | 0.722 | 12.935 |
| | IB-nnU-Net | **0.742*** | **11.125*** | 0.725 | 12.710 | 0.730 | 12.080 |
| 16 | SegResNet | 0.618 | 42.979 | 0.603 | 56.850 | 0.611 | 44.502 |
| | IB-SegResNet | 0.705 | 18.748 | 0.650 | 47.239 | 0.678 | 24.019 |
| | Attention U-Net | 0.582 | 60.322 | 0.241 | 196.081 | 0.589 | 50.391 |
| | IB-Att-U-Net | 0.622 | 38.066 | 0.461 | 113.534 | 0.617 | 41.565 |
| | TransUNet | 0.753 | 11.400 | 0.704 | 14.160 | 0.747 | 12.878 |
| | IB-TransUNet | 0.763 | 11.000 | 0.728 | 12.866 | 0.751 | 13.834 |
| | nnU-Net | 0.759 | 11.207 | 0.710 | 13.920 | 0.753 | 12.660 |
| | IB-nnU-Net | **0.796*** | **8.272*** | 0.760 | 9.675 | 0.783 | 10.403 |
| 24 | SegResNet | 0.645 | 30.436 | 0.532 | 56.045 | 0.594 | 56.915 |
| | IB-SegResNet | 0.768 | 9.315 | 0.596 | 39.574 | 0.723 | 22.853 |
| | Attention U-Net | 0.690 | 24.905 | 0.449 | 101.603 | 0.640 | 44.640 |
| | IB-Att-U-Net | 0.737 | 11.955 | 0.487 | 90.554 | 0.645 | 33.852 |
| | TransUNet | 0.800 | 9.150 | 0.791 | 8.624 | 0.797 | 8.939 |
| | IB-TransUNet | 0.802 | 9.050 | 0.791 | 8.566 | 0.793 | 8.420 |
| | nnU-Net | 0.803 | 8.938 | 0.794 | 8.424 | 0.800 | 8.732 |
| | IB-nnU-Net | **0.811*** | **8.863*** | 0.800 | 8.389 | 0.802 | 8.246 |

Table 3: Accuracy and robustness of U-Net variants on PROMISE-12 under no noise, Gaussian blur, and additive Gaussian noise.

**IB-nnU-Net Variants:** We conducted ablations to determine an effective IB-nnU-Net configuration (Figure 1). We explored adding IB layers after max-pooling (yielding only marginal improvements), introducing IBs symmetrically in encoder and decoder blocks, and inserting IB kernels in all encoder blocks (which resulted in overfitting). The most effective configuration places the IB block only in the second encoder block, with input taken before max-pooling and IB convolutions using stride 2. As shown in Table 1, this configuration yields the best trade-off between accuracy and robustness. We also experimented with kernel sizes $k \in \{3, 5, 7, 9\}$ and associated parameters $(r, \rho)$. A kernel size of $k = 3$ improved performance relative to nnU-Net but was less stable across tasks. Larger kernels increased memory and computation with no consistent benefit over $k = 5$, which is the default.

**Robustness on Small Training Subsets:** Tables 2 and 3 summarise the segmentation performance of all four original U-Nets, and their IB-extended variants on small subsets of MSD-hippocampus, MSD-prostate, MSD-spleen, and PROMISE-12. We report DSC and HD-95; SDC results are provided in the appendix. The IB extensions consistently outperform their corresponding baselines across nearly all settings. Among all variants, IB-nnU-Net achieves the highest accuracy and robustness, with pronounced gains for training sizes of 8 and 16. Figure 2 illustrates that IB-nnU-Net exhibits fewer spurious activations and more accurate delineation, especially in challenging PROMISE-12 and MSD-spleen scans. Figure 3 shows that IB kernels act as effective edge detectors, retaining sharper boundary information in early encoder blocks.

| Metric | Model | MSD hippo. ($n$=260) | MSD prostate ($n$=32) | PROMISE prostate ($n$=50) | MSD spleen ($n$=41) | AMOS 2022 ($n$=300) | PET $^{68}$Ga ($n$=68) | PET $^{18}$F ($n$=65) |
|---|---|---|---|---|---|---|---|---|
| DSC | nnU-Net | 0.909 | 0.882 | 0.890 | 0.966 | 0.886 | 0.711 | 0.768 |
| (↑) | IB-nnU-Net | **0.910** | **0.895** | **0.902** | **0.970** | **0.896** | **0.749** | **0.777** |
| HD-95 | nnU-Net | 1.068 | 1.980 | 1.735 | 1.640 | 1.932 | 12.458 | 10.764 |
| (↓) | IB-nnU-Net | **1.064** | **1.738** | **1.237** | **1.189** | **1.625** | **11.041** | **10.038** |

Table 4: Average 5-fold cross-validation accuracy of nnU-Net and IB-nnU-Net on full datasets. The standard deviations values are listed in the appendix.

| Training dataset | Testing dataset | DSC (↑) | | HD-95 (↓) | |
|---|---|---|---|---|---|
| | | nnU-Net | IB-nnU-Net | nnU-Net | IB-nnU-Net |
| PROMISE-12 | Prostate158 (Prostate) | 0.812 | **0.831** | 4.450 | **3.450** |
| MSD-prostate | Prostate158 (Prostate) | 0.826 | **0.845** | 4.050 | **3.150** |
| PROMISE-12 | PROSTATEx (Prostate) | 0.922 | **0.931** | 2.000 | **1.700** |
| MSD-spleen | AMOS-2022 (Spleen) | 0.927 | **0.940** | 1.800 | **1.350** |

Table 5: Out-of-distribution performance of nnU-Net and IB-nnU-Net.

**Robustness on Noisy Data:** PROMISE-12 is challenging due to anisotropic voxel spacing, heterogeneous acquisition protocols, and variable prostate appearance. To further probe robustness, we introduce Gaussian blur and additive Gaussian noise at test time. Table 3 reports performance across noise conditions and training subset sizes. In all settings, IB-nnU-Net matches or outperforms nnU-Net, with the largest gains in HD-95. The IB extensions confer similar robustness benefits to SegResNet, Attention U-Net, and TransUNet, indicating that IB kernels enhance resilience to acquisition artefacts and noise without sacrificing accuracy on clean data.

**Performance on Full Datasets:** We next evaluate nnU-Net and IB-nnU-Net using all available training volumes for MSD-heart, MSD-hippocampus, MSD-prostate, MSD-spleen, PROMISE-12, and AMOS-2022 via 5-fold cross-validation (Table 4). As expected for larger datasets, absolute DSC improvements are modest (Isensee et al., 2024), but IB-nnU-Net consistently matches or slightly surpasses nnU-Net, with more noticeable gains in HD-95, reflecting improved boundary localisation. On AMOS-2022, IB-nnU-Net achieves a mean DSC of 89.59% vs 88.64% for nnU-Net. On the official MSD test set (public leaderboard), IB-nnU-Net surpasses nnU-Net when both models are trained exclusively on the MSD training data. On PROMISE-12, IB-nnU-Net achieves a challenge score of 89.69 versus 89.65 for nnU-Net. Table 4 reports prostate tumour segmentation from PET images, and IB-nnU-Net improves over nnU-Net both tracers. This indicates that the proposed IB is also beneficial for challenging tumour segmentation in PET, characterised by noisy, low-resolution signals and heterogeneous tracer uptake.

**Out-of-Distribution Generalisation:** We examine cross-dataset generalisation without fine-tuning. Models trained on PROMISE-12 or MSD-prostate are evaluated on Prostate158 and PROSTATEx; models trained on MSD-spleen are evaluated on AMOS-2022 spleen cases (Table 5). IB-nnU-Net consistently outperforms nnU-Net across all out-of-distribution scenarios in both DSC and HD-95. These gains are practically relevant, as clinical deployment often involves datasets acquired with different scanners, protocols, or patient populations than those seen during training.

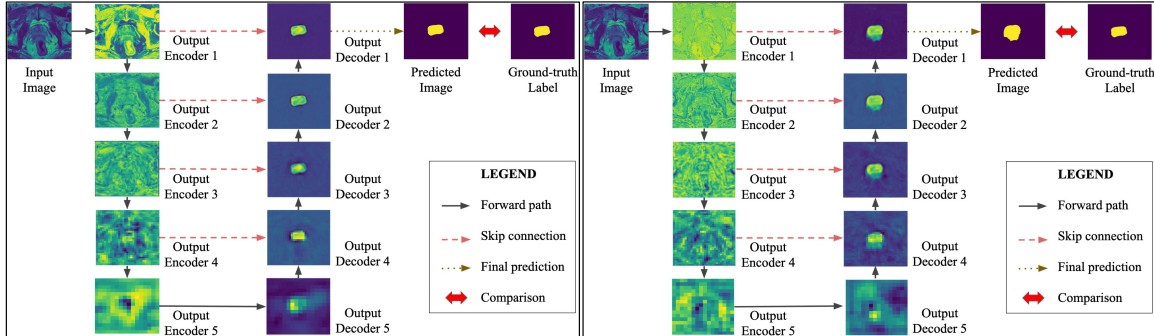

Figure 3: Feature maps from early encoder stages of IB-nnU-Net (left) and nnU-Net (right) trained for prostate segmentation. IB-nnU-Net retains sharper boundary information and reduces irrelevant activations.

| Comparison Type | Model Name/Setting | DSC (↑) | HD-95 (↓) | SDC (↑) |
|---|---|---|---|---|
| Different IB kernel parameters | $k=3$, $r=1$, $\rho=1/2$ | 0.640 | 29.318 | 0.639 |
| | $k=5$, $r=2$, $\rho=2/3$ | **0.665** | 27.522 | 0.651 |
| | $k=7$, $r=3$, $\rho=3/4$ | **0.665** | **27.510** | **0.652** |
| | $k=9$, $r=4$, $\rho=4/5$ | 0.660 | 28.591 | 0.650 |

Table 6: Ablation study on IB-nnU-Net design choices using a development subset of size 8 from MSD-Spleen. This is to determine whether adapting the parameters based on the same dataset produces different results compared to the PROMISE-12 subset.

## 5. Additional Analyses: Hyperparameter Transfer, Subset Sensitivity, and Learnable IB Kernels

In this section we analyse the generalisability of our fixed IB hyperparameters, the sensitivity of limited-data results to subset sampling, and the impact of allowing IB kernels to be learnable. We conducted three additional analyses on MSD-Spleen and improved variance reporting for cross-validation results.

**Dataset-specific hyperparameter adaptation on MSD-Spleen:** While our default IB configuration ($k=5, r=2, \rho=2/3$) was selected on a small PROMISE-12 development subset and then held fixed across all experiments, we additionally performed a dataset-specific hyperparameter ablation on a small MSD-Spleen development subset of size 8. Table 6 shows that the best-performing settings on MSD-Spleen are close to our default choice: $k=5$ yields the highest DSC, while $k=7$ achieves a marginally lower HD-95 and slightly higher SDC with essentially identical DSC. Overall, these results indicate that adapting the IB parameters to a different dataset does not change the main conclusion: the PROMISE-12-selected configuration transfers well, and the method is not brittle to modest changes in kernel size and associated parameters.

**Robustness to limited-data subset sampling:** For the limited-data experiments, subset construction can introduce sampling variance. To quantify this effect, we trained and evaluated all U-Net variants on a different randomly drawn 16-sample subset from MSD-Spleen using the same protocol and fixed IB kernels. The top block of Table 7 reports

| Size | Model Name | DSC(↑) | HD95(↓) | SDC(↑) | ΔDSC(↑) | ΔHD95(↓) | ΔSDC(↑) |
|------|-----------|--------|---------|--------|---------|----------|---------|
| | *Different 16-sample subset from MSD-Spleen trained with fixed IB kernels)* | | | | | | |
| 16 | SegResNet | 0.693 | 20.695 | 0.671 | −0.011 | +4.182 | −0.029 |
| | IB-SegResNet | 0.697 | 20.152 | 0.682 | −0.018 | +5.821 | −0.037 |
| | Att-U-Net | 0.696 | 18.098 | 0.681 | −0.032 | +5.334 | −0.036 |
| | IB-Att-U-Net | 0.698 | 17.302 | 0.687 | −0.041 | +5.323 | −0.038 |
| | Trans-U-Net | 0.706 | 14.120 | 0.674 | −0.039 | +3.720 | −0.062 |
| | IB-Trans-U-Net | 0.704 | 13.800 | 0.699 | −0.043 | +3.500 | −0.040 |
| | nnU-Net | 0.701 | 14.409 | 0.696 | −0.051 | +4.135 | −0.053 |
| | IB-nnU-Net | **0.725∗** | **12.102∗** | **0.706∗** | −0.031 | +1.921 | −0.056 |
| | *Learnable IB kernels using the original 16 samples from MSD-Spleen* | | | | | | |
| 16 | IB-SegResNet | 0.714 | 14.303 | 0.716 | −0.001 | −0.028 | −0.003 |
| | IB-Att-U-Net | 0.737 | 11.952 | 0.724 | −0.002 | −0.027 | −0.001 |
| | IB-Trans-U-Net | 0.749 | 10.223 | 0.738 | +0.002 | −0.077 | −0.001 |
| | IB-nnU-Net | **0.755** | **10.152** | **0.761** | −0.001 | −0.029 | −0.001 |

Table 7: Evaluating the IB-extended U-Nets with different a subset of samples, and training with IB kernels - no fixed parameters. For DSC and SDC metrics, increased delta signifies that the newer models performed better than the ones with original subsamples/fixed IB kernels, and vice-versa for HD-95 metric.

absolute metrics on this alternative subset alongside Δ values relative to the corresponding results obtained with the original subset. As expected, absolute performance shifts across architectures due to the changed subset; however, the overall pattern remains stable, with IB-extended variants remaining competitive and IB-nnU-Net continuing to provide strong accuracy/robustness. This supports that the observed improvements are not an artifact of a single particular subset draw.

**Fixed versus learnable IB kernels:** Here we evaluate on whether fixing the IB kernel weights is necessary, or whether allowing them to be learnable (starting from the same DoG initialization) provides meaningful gains. We therefore repeated training on the original 16-sample MSD-Spleen subset while allowing the IB kernel weights to update during training. The bottom block of Table 7 shows that learnable-IB performance is extremely close to the fixed-IB setting across backbones, with only minimal changes in DSC/SDC and HD-95. This suggests that the fixed kernels already capture most of the benefit of the inductive bias in this regime, supporting our design choice to keep the kernels fixed and parameter-neutral.

**Cross-validation variance reporting:** Finally, to improve statistical reporting for full-data experiments, we now explicitly note in the main cross-validation table (Table 4) that standard deviations across the five folds are provided in the appendix. This complements the mean performance values and helps assess variability across folds.

## 6. Discussion and Conclusion

We introduced two fixed 3D kernels inspired by on/off centre-surround pathways in the vertebrate retina and integrated them as inductive biases into 3D U-Net variants. These kernels act as complementary edge detectors with pre-computed weights and add no learnable parameters. When inserted into the second encoder block, they improve boundary representations and enhance robustness, particularly on small datasets and in out-of-distribution sce-

narios. Our experiments show that IB-extended U-Nets provide the strongest relative gains when training data are limited or when test data differ substantially from the training distribution, as reflected in improved HD-95 and SDC scores and qualitative visualisations (Figures 2 and 3). For large datasets with high-contrast structures, improvements over nnU-Net are smaller, consistent with performance saturation on well-curated benchmarks (Isensee et al., 2024).These benchmark experiments (Isensee et al., 2024) were conducted by the original authors of the nnU-Net framework, where they showcased the robustness of CNN-based architectures over recent ones such as transformers, mamba, etc. This is one of the main reasons we limited our model selections mainly to CNN-based (except TransUNet) architectures.

**Limitations.** IB kernels do not guarantee large gains in every setting. For large organs with clear boundaries and abundant training data ($> 50$), the inductive bias becomes less critical. Our ablations also show that naively placing IB kernels in all encoder blocks or symmetrically in encoder–decoder fashion can lead to overfitting or negligible benefits, underscoring the importance of the chosen configuration (second encoder block, $k = 5$).

**Future work.** Adapting the IB concept to transformer-based encoders - for example through attention-based analogues of centre-surround processing - could extend the benefits to a broader class of architectures. Allowing the IB kernel parameters to be lightly learned, but regularised toward the biologically motivated initialisation, may enable task-specific adaptation without sacrificing robustness. Systematic evaluation on additional modalities such as ultrasound and histopathology would further test generality.

The IB kernels introduce negligible computational overhead and retain the parameter count of the original U-Net variants, making them attractive as a drop-in modification for existing 3D segmentation pipelines. The improvements in boundary quality and robustness - particularly for small datasets, anisotropic acquisitions such as PROMISE-12, PET tumour segmentation, and cross-dataset transfer to Prostate158, PROSTATEx, and AMOS-2022 - suggest that such inductive biases are practically useful in clinical scenarios where collecting large amounts of labelled data is difficult.

In summary, equipping U-Net-style architectures with biologically inspired 3D IB kernels yields consistent robustness gains at virtually no parameter cost. IB-nnU-Net and related variants offer a simple yet effective approach to improving 3D medical image segmentation in clinically relevant regimes characterised by limited data and distribution shifts.

## Acknowledgments

This research was funded in part by the Austrian Science Fund (FWF) [grant number: I 6605], and the German Federal Ministry of Education and Research (BMBF) [grant number: 01KT2325] under the ERA-NET TRANSCAN-3 initiative project MATTO-GBM. The authors declare that there are no conflicts of interest.

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

## Appendix A. Additional Experimental details and Results

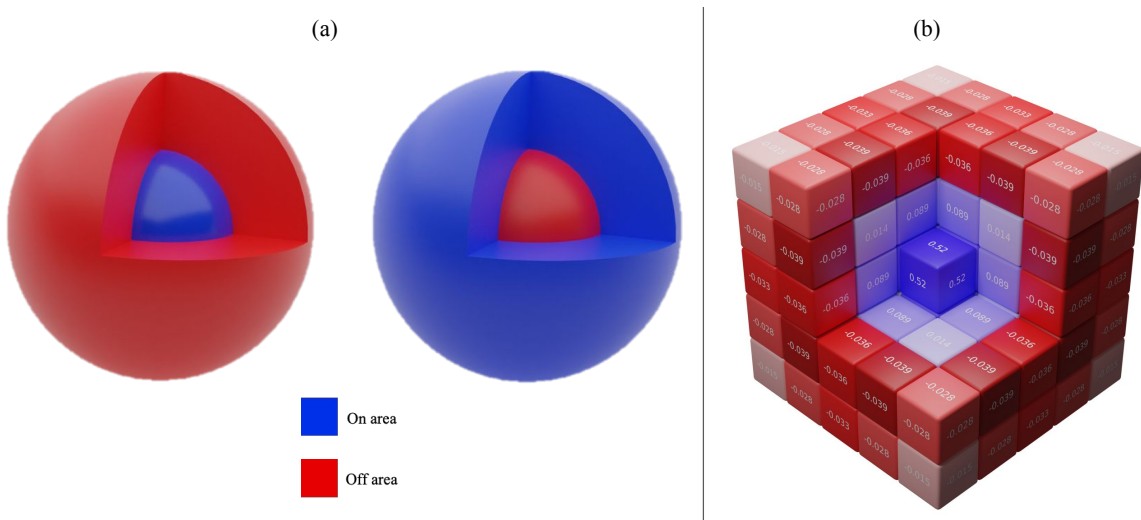

Figure 4: Geometrical representations of the IB kernels. (a) Spherical On and Off 3D centre-surround receptive fields. (b) The 3D IB-On cubic kernel. The 3D-IB Off cubic kernel is complementary: all its signs are inverted.

| Dataset | Patch Size | Batch |
|---|---|---|
| MSD-Heart | 160×160×64 | 2 |
| MSD-Hippocampus | 160×160×96 | 2 |
| MSD-Prostate | 160×160×64 | 2 |
| PROMISE-12 | 128×128×64 | 2 |
| MSD-Spleen | 192×160×64 | 2 |
| Private PET | 128×128×128 | 2 |

Table 8: Model training patch sizes and batch sizes.

| Training dataset | Testing dataset | SDC (↑) | |
|---|---|---|---|
| | | nnU-Net | IB-nnU-Net |
| PROMISE-12 | Prostate158 (Target: Prostate) | $0.822 \pm 0.133$ | $\mathbf{0.841 \pm 0.080}$ |
| MSD-prostate | Prostate158 (Target: Prostate) | $0.836 \pm 0.093$ | $\mathbf{0.855 \pm 0.069}$ |
| PROMISE-12 | PROSTATEx (Target: Prostate) | $0.932 \pm 0.050$ | $\mathbf{0.941 \pm 0.044}$ |
| MSD-spleen | AMOS-2022 (Target: Spleen) | $0.933 \pm 0.130$ | $\mathbf{0.946 \pm 0.081}$ |

Table 9: SDC results of the nnU-Net and IB-nnU-Net on out-of-distribution samples.

| Size | Model Name | MSD-Hippocampus SDC(↑) | MSD-Prostate SDC(↑) | MSD-Spleen SDC(↑) |
|---|---|---|---|---|
| 8 | SegResNet | 0.568 | 0.523 | 0.602 |
| | IB-SegResNet | 0.585 | 0.605 | 0.607 |
| | Attention U-Net | 0.630 | 0.490 | 0.613 |
| | IB-Att-U-Net | 0.636 | 0.578 | 0.620 |
| | Trans-U-Net | 0.642 | 0.578 | 0.638 |
| | IB-Trans-U-Net | 0.645 | 0.665 | 0.667 |
| | nnU-Net | 0.643 | 0.614 | 0.655 |
| | IB-nnU-Net | **0.655**∗ | **0.699**∗ | **0.658**∗ |
| 16 | SegResNet | 0.680 | 0.699 | 0.700 |
| | IB-SegResNet | 0.694 | 0.740 | 0.719 |
| | Attention U-Net | 0.733 | 0.684 | 0.717 |
| | IB-Att-U-Net | 0.745 | 0.756 | 0.725 |
| | Trans-U-Net | 0.733 | 0.751 | 0.736 |
| | IB-Trans-U-Net | 0.756 | 0.796 | 0.739 |
| | nnU-Net | 0.751 | 0.758 | 0.749 |
| | IB-nnU-Net | **0.800**∗ | **0.797**∗ | **0.762**∗ |
| 24 | SegResNet | 0.792 | 0.799 | 0.822 |
| | IB-SegResNet | 0.813 | 0.809 | 0.812 |
| | Attention U-Net | 0.838 | 0.799 | 0.828 |
| | IB-Att-U-Net | 0.851 | 0.823 | 0.833 |
| | Trans-U-Net | 0.831 | 0.813 | 0.863 |
| | IB-Trans-U-Net | 0.839 | 0.818 | 0.866 |
| | nnU-Net | 0.852 | 0.820 | 0.856 |
| | IB-nnU-Net | **0.864**∗ | **0.829**∗ | **0.877**∗ |

Table 10: SDC results of U-Net variants on MSD-Hippocampus, MSD-Prostate, and MSD-Spleen datasets.

| Size | Model Name | No-Noise SDC(↑) | Gaussian Blur SDC(↑) | Random Gaussian SDC(↑) |
|---|---|---|---|---|
| 8 | SegResNet | 0.551 | 0.386 | 0.566 |
| | IB-SegResNet | 0.579 | 0.494 | 0.594 |
| | Attention U-Net | 0.455 | 0.461 | 0.445 |
| | IB-Att-U-Net | 0.638 | 0.526 | 0.615 |
| | Trans-U-Net | 0.655 | 0.563 | 0.666 |
| | IB-Trans-U-Net | 0.658 | 0.585 | 0.646 |
| | nnU-Net | 0.647 | 0.556 | 0.658 |
| | IB-nnU-Net | **0.670**∗ | 0.596 | 0.658 |
| 16 | SegResNet | 0.573 | 0.472 | 0.507 |
| | IB-SegResNet | 0.597 | 0.608 | 0.565 |
| | Attention U-Net | 0.608 | 0.322 | 0.642 |
| | IB-Att-U-Net | 0.649 | 0.482 | 0.689 |
| | Trans-U-Net | 0.699 | 0.664 | 0.685 |
| | IB-Trans-U-Net | 0.701 | 0.663 | 0.645 |
| | nnU-Net | 0.713 | 0.677 | 0.699 |
| | IB-nnU-Net | **0.762**∗ | 0.721 | 0.701 |
| 24 | SegResNet | 0.665 | 0.543 | 0.662 |
| | IB-SegResNet | 0.744 | 0.661 | 0.683 |
| | Attention U-Net | 0.625 | 0.475 | 0.604 |
| | IB-Att-U-Net | 0.652 | 0.646 | 0.638 |
| | Trans-U-Net | 0.766 | 0.714 | 0.739 |
| | IB-Trans-U-Net | 0.771 | 0.725 | 0.734 |
| | nnU-Net | 0.794 | 0.740 | 0.766 |
| | IB-nnU-Net | **0.810**∗ | 0.762 | 0.771 |

Table 11: SDC results of U-Net variants on PROMISE-12 under No-Noise, Gaussian Blur Noise, and Random Gaussian Noise.

| Metric | Model Name | MSD-hippocampus (260) | MSD-prostate (32) | PROMISE-12 (50) | MSD-spleen (41) |
|---|---|---|---|---|---|
| SDC (↑) | nnU-Net | $0.986 \pm 0.016$ | $0.876 \pm 0.117$ | $0.910 \pm 0.121$ | $0.976 \pm 0.052$ |
| | IB-nnU-Net | $\mathbf{0.986 \pm 0.015}$ | $\mathbf{0.879 \pm 0.078}$ | $\mathbf{0.920 \pm 0.093}$ | $\mathbf{0.982 \pm 0.026}$ |

Table 12: SDC results of U-Net variants and their IB extensions on the full datasets.

| Metric (SD) | Model | MSD hippo. ($n$=260) | MSD prostate ($n$=32) | PROMISE prostate ($n$=50) | MSD spleen ($n$=41) | AMOS 2022 ($n$=300) | PET $^{68}$Ga ($n$=68) | PET $^{18}$F ($n$=65) |
|---|---|---|---|---|---|---|---|---|
| DSC SD (↑) | nnU-Net | 0.028 | 0.118 | 0.101 | 0.044 | 0.070 | 0.057 | 0.061 |
| | IB-nnU-Net | 0.028 | 0.042 | 0.036 | 0.017 | 0.032 | 0.026 | 0.027 |
| HD-95 SD (↓) | nnU-Net | 0.177 | 1.203 | 0.565 | 4.708 | 0.730 | 4.963 | 4.512 |
| | IB-nnU-Net | 0.166 | 1.122 | 0.556 | 1.914 | 0.629 | 4.057 | 3.505 |

Table 13: Fold-wise standard deviation (SD) across 5-fold cross-validation for the full-dataset results in Table 4.

| Training dataset | Testing dataset | DSC SD (↑) | | HD-95 SD (↓) | |
|---|---|---|---|---|---|
| | | nnU-Net | IB-nnU-Net | nnU-Net | IB-nnU-Net |
| PROMISE-12 | Prostate158 (Prostate) | 0.131 | **0.079** | 1.873 | **1.772** |
| MSD-prostate | Prostate158 (Prostate) | 0.092 | **0.068** | 1.372 | **1.172** |
| PROMISE-12 | PROSTATEx (Prostate) | 0.049 | **0.044** | 0.479 | **0.229** |
| MSD-spleen | AMOS-2022 (Spleen) | 0.129 | **0.081** | 0.653 | **0.624** |

Table 14: Standard deviation (SD) estimates for the out-of-distribution metrics in Table 5.

