# OpenReview forum: "Learning Robust Medical Image Segmentation with Inductive Bias"
_MIDL.io/2026/Conference — MIDL 2026 Poster_

### Official Review · Reviewer_FPVW · 2026-01-04

**Confidence:** 3
**Preliminary Rating:** 4
**Final Rating:** 4

**Summary:**

This paper proposes integrating biologically inspired inductive biases into 3D U-Net architectures to improve robustness in medical image segmentation. Two fixed 3D residual components representing on- and off-center-surround convolutions are inserted into the second encoder block to act as complementary edge detectors. Extensive experiments across multiple datasets and architectures demonstrate that this modification enhances performance particularly in small-data regimes and out-of-distribution scenarios without adding learnable parameters.

**Strengths:**

- Integrating fixed, biologically plausible filters into deep learning models is a refreshing approach that effectively bridges neuroscience and computer vision without overcomplicating the architecture.
- Experimental validation is exceptionally rigorous, covering four different baseline architectures (including nnU-Net and TransUNet) and multiple diverse datasets ranging from MRI to CT and PET.
- Analysis of small-data regimes and out-of-distribution generalization addresses a critical bottleneck in medical imaging where labeled data is often scarce.
- Proposed method adds virtually no computational overhead and zero additional learnable parameters, making it highly practical for clinical deployment.
- Ablation studies regarding filter placement and kernel shape provide valuable intuition into why the specific design choices work, specifically the superiority of spherical over cylindrical kernels.

**Weaknesses:**

- Improvements on the full datasets are relatively marginal compared to the baseline nnU-Net, suggesting the benefit diminishes significantly as data abundance increases.
- Fixing the kernel parameters prohibits the model from fine-tuning these filters for specific organ characteristics or image contrasts, which might limit peak performance compared to a learnable initialization.
- Justification for inserting the block specifically in the second encoder layer is primarily empirical, and a more theoretical grounding for why this specific resolution level requires edge enhancement would be beneficial.
- Mathematical formulation of the 3D kernels is a straightforward extension of existing 2D work, limiting the technical novelty strictly to the application domain and implementation details.
- Visualizations of the feature maps show sharper edges, but a deeper feature space analysis is missing to confirm if the network truly utilizes these pathways as intended or simply compensates for the fixed weights elsewhere.

**Detailed Comments:**

- Please clarify if the normalization constant $c$ in Equation 3 requires adjustment across datasets with vastly different intensity distributions (e.g., CT vs. MRI).
- Consider adding a comparison where the IB kernel weights are initialized as proposed but allowed to update during training to verify the claim that fixed weights are superior or sufficient.
- Figure 1 is informative, but the connection between "On Convolved Output" and the downstream concatenation could be visually clearer regarding tensor dimensions.
- Standard deviation values in Tables 9 and 10 would help assess statistical significance more broadly than just the p-values mentioned in the text.
- Section 3.3.1 mentions "Naively extending this to 3D... is suboptimal," which is a strong claim that would benefit from a small quantitative comparison in the appendix if space permits.

**Justification Of Final Rating:**

The rebuttal addressed most of my concerns: fixed kernels suffice, parameters generalize across datasets. Significant small-data/OOD improvements justify this zero-overhead biological prior despite marginal large-dataset gains.

**Justification Of The Preliminary Rating:**

- This paper presents a solid, well-executed study on enhancing U-Net robustness using fixed biological priors.
- While the technical innovation of extending DoG filters to 3D is modest, the comprehensive evaluation across multiple architectures and data regimes provides convincing evidence of its utility.
- Focus on small-data and OOD performance is highly relevant to the MIDL community where labeled data is the primary constraint.
- Simplicity of the method is a virtue here, as it allows for easy integration into existing pipelines without resource overhead.
- Marginal gains on large datasets prevent a higher rating, but the significant boost in constrained settings warrants acceptance.

**Questions To Address In The Rebuttal:**

- Why was the decision made to keep the parameters fixed rather than using the biological initialization as a starting point for learnable weights, given that deep learning usually benefits from end-to-end optimization?
- Can you elaborate on the sensitivity of the model to the hyperparameter $k$ (kernel size) when voxel spacings vary significantly between datasets like PROMISE-12 and MSD-Spleen?
- Did you observe any negative transfer or degradation in performance when applying this specific inductive bias to imaging modalities with different noise profiles, such as ultrasound, if such experiments were considered?

---

> ### Author Response · Authors · 2026-01-25
>
> We thank the reviewer for the supportive and detailed review and for emphasizing the practical relevance and strong empirical evaluation.
>
> 1. Fixed vs learnable kernels: We keep kernels fixed to isolate the effect of the inductive bias under a strict parameter budget: IB-nnU-Net has exactly the same number of trainable parameters as nnU-Net. To address the reviewer’s suggestion, we added a learnable-kernel variant (initialized with the same DoG shape). Table 7 (bottom block) shows that allowing kernels to update yields only minimal changes relative to the fixed-kernel version, suggesting fixed kernels already approximate what the network would otherwise learn and supporting our design choice.
>
> 2. Sensitivity to voxel spacing and kernel size: nnU-Net resamples each dataset to a task-specific target spacing. The IB operates in this resampled space, so its effective physical scale follows nnU-Net’s preprocessing. Our kernel-size ablation experiment shows  k = 5  is a strong default. To further evaluate dataset dependence, we also added an MSD-Spleen dev-set ablation (Table 6), which again supports that the default  k = 5 is near-optimal and that performance is not overly sensitive within  k ∈ { 3 , 5 , 7 , 9 } k∈{3,5,7,9}.
>
> 3. Modalities with different noise profiles (e.g., ultrasound): We did not evaluate ultrasound in this work. However, we include MRI, CT, and PSMA-PET (two tracers), spanning substantially different noise and contrast profiles, and we do not observe negative transfer. We explicitly note ultrasound/histopathology evaluation as future work in the revised manuscript.
>
> Additional concern points
>
> (1) “Naively extending to 3D is suboptimal”: We already include a direct comparison between cylindrical and spherical variants in the main ablation, where cylindrical kernels can underperform nnU-Net, while spherical kernels yield consistent improvements. We also clarify the wording to emphasize that the issue is anisotropic bias from cylindrical stacks rather than an abstract claim.
>
> (2) Standard deviations: We now provide standard deviations across folds in the appendix, referenced from Table 4 and 5.
>
> (3) Normalization constant / intensity distributions (CT vs MRI):
> The IB kernels are constructed to reflect balanced centre–surround responses and are applied within nnU-Net’s standardized preprocessing/normalization pipeline. We clarify in the revised manuscript that we do not require modality-specific kernel re-scaling in our experiments.
>
> (4) Feature-space use beyond visualizations:  We provide attention-map and feature-map visualizations (figures 1 and 2) that show sharper boundary emphasis and fewer spurious activations. Additionally, the learnable-IB experiment (Table 7, bottom block) indicates the network does not need to substantially deviate from the biologically shaped filters to reach similar performance, which is consistent with the intended role of the IB as a stable edge/contrast prior.
>
> For large datasets with high-contrast structures, improvements over nnU-Net are smaller, consistent with performance saturation on well-curated benchmarks. These benchmark experiments were conducted by the original authors of the nnU-Net framework, where they showcased the robustness of CNN-based architectures over recent ones such as transformers, mamba, etc. This is one of the main reasons we limited our model selections mainly to CNN-based (except TransUNet) architectures.
> Reference: Isensee, F., Wald, T., Ulrich, C., Baumgartner, M., Roy, S., Maier-Hein, K., & Jaeger, P. F. (2024, October). nnu-net revisited: A call for rigorous validation in 3d medical image segmentation. In International Conference on Medical Image Computing and Computer-Assisted Intervention (pp. 488-498). Cham: Springer Nature Switzerland.
>
> We are also addressed other required changes made the reviewer in the revised manuscript.

---

### Official Review · Reviewer_NSsv · 2026-01-07

**Confidence:** 5
**Preliminary Rating:** 4
**Final Rating:** 4

**Summary:**

This study introduces a novel approach to enhance the robustness of 3D medical image segmentation models by integrating biologically inspired inductive bias filters, modelled after the on-center/off-center receptive fields of vertebrate retinas. The proposed filters, designed as 3D difference-of-Gaussian kernels, are embedded into the second encoder block of U-Net architectures without introducing additional trainable parameters. Experimental results across multiple medical datasets, including MSD, PROMISE-12, and AMOS-2022, demonstrate that this method outperforms baseline models, particularly in low-data and out-of-distribution scenarios, as shown by improvements in Dice similarity coefficient and Hausdorff distance. The approach seems to maintain computational efficiency, with negligible increases in VRAM usage and training time.

**Strengths:**

The strengths of the approach are as follows:
- The simplicity of implementation facilitates its adoption and integration into existing workflows without requiring extensive modifications.
- The quality of the results, along with the thorough comparisons and ablation studies, provides robust evidence of the method's effectiveness and reliability across various scenarios.
- The computational efficiency (no additional parameters are introduced) ensures that the approach remains practical and scalable.

**Weaknesses:**

The article does not exhibit any major weaknesses. The primary limitations of the approach are openly acknowledged by the authors in the discussion section, specifically the restriction to U-Net architectures and the use of fixed parameters for the inductive bias filters (which has a computational advantage), the diminishing return as datasets get bigger. The improvements over other approaches, while consistent, are sometimes modest in magnitude and not always significant from a statistical point of view.

**Detailed Comments:**

The paper is clear and well-written. The proposed approach is presented in a manner that is straightforward to understand and follow.
The paper does not explicitly detail the baseline used for the Wilcoxon signed-rank test when assessing the improvements of the IB-nnU-Net. Typically, in such studies, the baseline would be the best-performing existing model (e.g., nnU-Net) on the same dataset and under the same evaluation conditions. Is it the case?
As far as I understood, the evaluation of performance with limited training data is conducted using a single split for creating subsets of sizes 8, 16, and 24. This approach could be sensitive to sampling effects, as the performance may vary significantly depending on which samples are included in the subset. Could the evaluation be done using several splits?
For the evaluation on full datasets, a 5-fold cross-validation is employed, which is a robust approach. However, the paper could enhance the reporting of results by including standard deviation or CI of the performance metrics across the folds.

**Justification Of Final Rating:**

I acknowledge that the authors’ revisions have improved the paper. However, while the amendments represent a clear enhancement, they do not fully address the concerns necessary to warrant a strong recommendation for acceptance.

**Justification Of The Preliminary Rating:**

My preliminary evaluation recommends a weak acceptance to this study, Indeed the proposed approach, which integrates biologically inspired inductive bias filters into medical image segmentation models, demonstrates clear methodological innovation and achieves consistent improvements over established state-of-the-art models, such as nnU-Net. However, the magnitude of these improvements, while clear, remains relatively modest in comparison to certain baseline models, particularly in scenarios where existing methods already perform robustly, or significant in scenarios where sample effect may be strong (limited data).
The study’s findings would benefit from further validation across a broader spectrum of datasets, contexts, and architectural frameworks to strenghten their generalisability. While the authors transparently acknowledge these limitations, to me, the approach’s full potential has yet to be fully explored.

**Questions To Address In The Rebuttal:**

I find the paper interesting, it could be improved a bit but I am not sure it would change my rating.

---

> ### Author Response · Authors · 2026-01-25
>
> We thank the reviewer for the very positive review and for pointing out useful clarifications.
>
> Baseline and Wilcoxon signed-rank test:
> Yes, the Wilcoxon signed-rank test is computed pairwise against the corresponding baseline trained under identical conditions (same splits, same nnU-Net plans, same loss, optimiser, schedule, augmentations, and evaluation). We clarify this explicitly in the revised manuscript.
>
> Limited-data subsets and sensitivity to sampling:
> We agree that using a single subset draw can be sensitive to sampling. To address this concern directly, we added an experiment using a different 16-sample subset from MSD-Spleen under the same protocol. Table 7 (top block) reports absolute metrics for that new subset and the deltas relative to the original subset results. While absolute performance shifts (as expected under different sampling), IB-extended models remain competitive, and IB-nnU-Net remains the best performer among the compared backbones on that new subset. This supports that the observed robustness benefit is not a one-off artifact of a particular subset draw.
>
> Reporting variance/uncertainty in 5-fold cross-validation: We agree and have now included a table that explicitly notes that standard deviations are provided in the appendix, strengthening the statistical reporting beyond mean-only results.

---

### Official Review · Reviewer_X1LS · 2026-01-12

**Confidence:** 4
**Preliminary Rating:** 4

**Summary:**

The paper presents an inductive bias spherical kernel augmenting nn-UNets. Two 3D kernels determined with pre-set hyperparameters are inserted in the second encoder block of a 3D nn-UNet, representing on- and off-center-surround convolutions. The effectiveness of this approach is shown across various challenging datasets of various organs (the hippocampus, spleen, and prostate).

**Strengths:**

The motivation for spherical kernels is well-explained and there is ample experimentation results. Additionally, I appreciated the breadth of ablations, especially for the place of kernel insertion in the nnUNet architecture.

**Weaknesses:**

More intuition for hyperparameter choice would be nice as it is unclear if the kernel will generalise beyond these small datasets, challenging datasets. I also wonder if fixing these hyperparameters is the best approach as I imagine flexibility in learnable parameters is helpful if method is applied to other datasets.

**Detailed Comments:**

Prostate images, for instance, may have blurry surroundings and irregular shapes that makes it not directly suitable for this kernel/hyperparameter-determination, are there concerns regarding this design choice? Additionally, more justification for "choice of hyperparameters based on of spherical 3D kernels" is appreciated as it is not immediately obvious. Moreover, nn-UNet auto configures network to optimise for performance given dataset. This variability makes it difficult to compare performance gain attributed to IB Kernels across training datasets. I wonder if there is a good way to assess this. Furthermore, would the nnUNet configuration affect the best place to put the kernels?

**Justification Of The Preliminary Rating:**

While spherical kernels and DoG are long standing concepts and prior works explain this idea in the 2D case, the paper extrapolates to 3D, justifying the DoG spherical kernel construction and showing novelty in its inclusion in the popular nn-UNet architecture. There are meaningful performance gains across a range of datasets with little compute overhead. More justification on hyperparameter choice would be appreciated.

**Questions To Address In The Rebuttal:**

1) Are the validations of IB hyperparameters on the small development subset from PROMISE-12 sufficient? Could potentially adapting the parameters based on dataset yield better results?
2) Does this extend well beyond the hippocampus, prostate, and spleen? How spherical in shape does the organ to be segmented need to be for this to work well?
3) Aside from computation ease, what are the benefits of precomputing this kernel? Computationally, would it be very expensive to optimize this kernel?
4) Training employed augmentation. Will distortion augmentations (if applied) affect the effectiveness of the kernel?

---

> ### Author Response · Authors · 2026-01-25
>
> We thank the reviewer for the positive assessment, particularly for highlighting the motivation for spherical kernels and the breadth of ablations.
>
> 1) Are the validations of IB hyperparameters on PROMISE-12 (n=8) sufficient? Would adapting per dataset help?
> Our goal is not to find dataset-specific optimal hyperparameters, but to test whether a single, biologically motivated 3D IB configuration generalises across tasks and modalities. We selected the default setting  ( k = 5 , r = 2 , ρ = 2 / 3 ) (k=5,r=2,ρ=2/3) on a small PROMISE-12 development subset (n=8) and then kept these parameters fixed across all datasets. To directly address the reviewer’s question about dataset-specific adaptation, we added an additional hyperparameter ablation on a development subset of MSD-Spleen (n=8). Table 6 shows that the best-performing settings on this dataset are very close to the PROMISE-tuned default:  k = 5 remains best for DSC (and competitive for HD-95/SDC), while  k = 7 yields nearly identical outcomes. This supports that the PROMISE-based choice is not brittle and that dataset-specific tuning does not materially change conclusions, at least within the tested parameter range.
>
> 2) Does this extend beyond hippocampus/prostate/spleen? Do organs need to be spherical?
> The kernels are “spherical” only in terms of isotropic 3D filter support, which avoids anisotropic bias that arises when stacking 2D kernels into cylindrical 3D filters. They do not assume the target anatomy is globally spherical. They operate on local centre–surround contrast and emphasize boundaries/edges and blob-like local structures. Empirically, we apply the same IB configuration across compact but non-spherical structures (hippocampus, prostate), large irregular organs (spleen and multiple abdominal organs in AMOS-2022), and highly irregular PSMA-PET tumours with heterogeneous uptake. Across these settings, the IB improves robustness metrics (HD-95/SDC) and often DSC, indicating the benefit is not restricted to “spherical organs.”
>
> 3) Why precompute the kernel? Would optimizing it be expensive?
> The primary motivation for fixed kernels is conceptual: we incorporate a known, interpretable inductive bias without increasing the number of trainable parameters and without adding new optimisation degrees of freedom. This keeps IB-nnU-Net parameter-matched to nnU-Net and isolates the effect of the inductive bias. We also tested the reviewer’s suggestion by allowing the kernels to be learnable (initialized from the same DoG shape). As reported in Table 7 (bottom block), performance is extremely close to the fixed-kernel setting, suggesting that the fixed IB already captures most of the benefit.
>
> 4) Will augmentation/distortion affect the IB effectiveness?
> All models are trained with the standard nnU-Net augmentation pipeline under identical settings, so reported results already reflect training with augmentation. Additionally, we explicitly evaluate robustness to test-time Gaussian blur and additive Gaussian noise on PROMISE-12, where IB-nnU-Net matches or outperforms nnU-Net, especially in HD-95. This indicates the IB remains effective under typical distortions and acquisition artifacts.
>
> Additional comment: nnU-Net auto-configuration and kernel placement
> For each dataset, nnU-Net first derives its plans (depth, patch size, target spacing, etc.). IB-nnU-Net uses the same plan and identical training pipeline; the only change is replacing the second encoder block with the IB block. Our placement ablations show that inserting the IB in Encoder 2 provides the best stability–performance trade-off even as nnU-Net plans vary across datasets. Intuitively, Encoder 2 is the first downsampled scale where boundaries remain spatially local but features become less dominated by raw intensity noise. This suggests that the fixed kernels already approximate the shapes that the network would otherwise learn. This is consistent with recent work showing that depthwise convolutional filters in CNNs tend to converge to biologically plausible receptive-field shapes (Babaiee et al. 2024).
>
> Reference: Babaiee, Z., Kiasari, P. M., Rus, D., & Grosu, R. (2024). Neural echos: Depthwise convolutional filters replicate biological receptive fields. In Proceedings of the IEEE/CVF Winter Conference on Applications of Computer Vision (pp. 8216-8225).

---

> > ### Comment · Reviewer_X1LS · 2026-02-02
> > **Final Rating**
> >
> > I appreciate the authors' thorough and detailed response to my questions; all of which were very helpful in clearing my concerns. I will keep my previous review of weak accept.

---

### Author Rebuttal · Authors · 2026-01-25

**Rebuttal:**

We are grateful to all the reviewer's encouraging comments and constructive feedback. We have addressed each reviewer's concerns in the comment section of respective reviewers, with changes highlighted in RED colored text in our revised manuscript .

We are attaching the revised rebuttal manuscript as the supporting material here.

**Supporting Material:**

/attachment/4b39c7d925658d4a506694d65da0cbc91c2de449.pdf

---

### Meta-Review · Area_Chair_XzwB · 2026-02-08

**Recommendation:** Accept (Poster)
**Confidence:** 4

**Metareview:**

The paper has three clear weak accept (before and after rebuttal). The authors answered all the queries raised by the reviewers during the rebuttal period. All the reviewers appreciated the methodology, ablation study, and multiple baselines in the paper. Considering everything, I am recommending acceptance of the paper.

---

### Decision · Program_Chairs · 2026-02-13

Accept (Poster)